# Pathophysiology of Cardiac Injury in COVID-19 Patients with Acute Ischaemic Stroke: What Do We Know So Far?—A Review of the Current Literature

**DOI:** 10.3390/life12010075

**Published:** 2022-01-06

**Authors:** Daniela Schoene, Luiz G. Schnekenberg, Lars-Peder Pallesen, Jessica Barlinn, Volker Puetz, Kristian Barlinn, Timo Siepmann

**Affiliations:** Department of Neurology, University Hospital Carl Gustav Carus, Technische Universitaet Dresden, 01307 Dresden, Germany; Luiz.Schnekenberg@ukdd.de (L.G.S.); lars-peder.pallesen@ukdd.de (L.-P.P.); jessica.barlinn@ukdd.de (J.B.); Volker.Puetz@ukdd.de (V.P.); Kristian.Barlinn@ukdd.de (K.B.); timo.siepmann@ukdd.de (T.S.)

**Keywords:** acute ischemic stroke, cardiac injury, COVID-19, SARS-CoV-2 infection, stroke pathogenesis, endotheliopathy, endothelial dysfunction

## Abstract

With the onset of the COVID-19 pandemic, it became apparent that, in addition to pulmonary infection, extrapulmonary manifestations such as cardiac injury and acute cerebrovascular events are frequent in patients infected with SARS-CoV-2, worsening clinical outcome. We reviewed the current literature on the pathophysiology of cardiac injury and its association with acute ischaemic stroke. Several hypotheses on heart and brain axis pathology in the context of stroke related to COVID-19 were identified. Taken together, a combination of disease-related coagulopathy and systemic inflammation might cause endothelial damage and microvascular thrombosis, which in turn leads to structural myocardial damage. Cardiac complications of this damage such as tachyarrhythmia, myocardial infarction or cardiomyopathy, together with changes in hemodynamics and the coagulation system, may play a causal role in the increased stroke risk observed in COVID-19 patients. These hypotheses are supported by a growing body of evidence, but further research is necessary to fully understand the underlying pathophysiology and allow for the design of cardioprotective and neuroprotective strategies in this at risk population.

## 1. Introduction

Acute ischemic stroke (AIS) is an emerging vascular complication in patients with COVID-19 with reports on incidence ranging between 1 and 6% of hospitalized patients [1]. Among pathophysiological causes of AIS in COVID-19, SARS-CoV-2 related cardiac injury, mediated by a thrombotic and inflammatory milieu, seems to play a major role. Frequently reported manifestations of cardiac injury in COVID-19 patients comprise right ventricular dysfunction, heart failure, circulatory shock, myocarditis, cardiomyopathy, arrhythmia and/or thromboembolic events [2,3,4]. Cardiac injury occurs in up to 46.3% of COVID-19 patients and may be even more frequent in SARS-CoV-2 positive AIS patients [5]. Moreover, cardiac structural damage related to SARS-CoV-2 is associated with increased mortality, as shown in a cohort study in Wuhan (42 of 82 [51.2%] vs. 15 of 334 [4.5%]; *p* < 0.001) [6]. Hence, biomarkers of cardiac injury might be helpful when designing strategies of intensified monitoring of COVID-19 patients to prevent cardiovascular and cerebrovascular complications such as AIS.

The disease progression of COVID-19 can be divided into three, possibly overlapping, phases: early infection (stage I), pulmonary phase (stage II) and hyperinflammation (stage III) [7]. In an initial infection phase, the primary target of the virus is the lung parenchyma. The innate immune defense is immediately activated. Consequently, an inflammatory reactive process occurs resulting in damage to the vessel walls, which leads to vasodilatation and endothelial permeability. This leads to pulmonary restriction and subsequently to further hypoxemia, forming a vicious circle and finally leading to increased cardiovascular stress [7]. In the case of systemic (hyper-) inflammation, the damage to organs (to multiple organ failure) as the myocardium comes into focus [7]. Vascular involvement starts in stage II and is characterized by endothelial damage due to inflammatory reaction. In addition, in stage III, cardiac stress occurs due to respiratory failure and mechanisms of cardiac injury. These mechanisms include viral infiltration into myocardial tissue as well as secondary reactions to hypoxemia and cardiac inflammation, subsequently leading to cardiac complications [7]. Biomarkers as indicators of (hyper-)inflammation include IL (interleukin)-6, IL-2, IL-7, TNF (tumor necrosis factor)-α, IFN (interferon)-γ IP (inducible protein)-10, MCP (monocyte chemoattractant protein)-1, MIP (macrophage inflammatory protein)-1α, G-CSF (granulocyte-colony stimulating factor), CRP (C-reactive protein), procalcitonin and ferritin. Elevations of these markers are associated with an increase in mortality [7]. The results of a cohort study with 84 patients diagnosed with COVID-19 from Wuhan, China, demonstrated that the level of cardiac enzymes, as well as the abnormalities in the ECG, correlate positively with the level of inflammation values, in particular CRP (*p* = 0.004) and procalcitonin (*p* = 0.012) [8]. It can therefore be assumed that common, overlapping and interacting mechanisms of heart and brain axis pathology are associated with COVID-19 disease determining the severity of the disease. 

We performed a narrative review of the current literature on the association of cardiac injury and acute ischemic stroke in patients with COVID-19 with a focus on pathophysiological mechanisms and potential biomarkers of cardiac involvement in this population at risk.

## 2. Search Strategy

A literature search was performed by using Medline via the PubMed interface and the Cochrane Library (Cochrane Database of Systematic Reviews, Cochrane Central Register of Controlled Trials [CENTRAL] and Cochrane Methodology Register). Furthermore, we screened the reference lists of the included studies. We only included literature in the English language. The literature search was performed between 2 April 2021 and 22 April 2021. We included randomized and non-randomized clinical trials, prospective and retrospective observational studies as well as previous reviews and meta-analyses. We applied the Medical Subject Headings “COVID-19” and “COVID” in combination with each of the following terms; “cardiac injury,” “cardiac” “cardial” “cardiovascular,” “Troponin” “ischemic stroke” “stroke” using the Boolean operators “AND” as well as “OR” and their combinations. The initial search provided 19,090 results, which were screened according to content aspects related to the topic of the review and evaluated according to relevance. Explicit exclusion criteria were not defined. A selection bias cannot be ruled out.

## 3. Pathophysiology and Clinical Manifestations of Cardiac Injury in COVID-19

The systemic inflammatory virus disease COVID-19 is associated with impaired cardiac function and can subsequently lead to persistent structural myocardial damage [9]. According to the European society of cardiology (ESC), myocardial injury is defined as “detection of elevated cardiac troponin (cTn) values above the 99th percentile upper reference limit“. The injury is considered acute “if there is a rise and/or fall or cTn values”. This definition applies to a variety of situations, e.g., after a coronary procedural intervention [10]. In COVID-19, virus infection leads to systemic inflammation. This inflammatory milieu causes cardiac structural damage mediated by inflammatory cytokines. In rare cases, this leads to fulminant myocarditis even without respiratory involvement [11]. Moreover, coronary vessel inflammation leads to cardiac ischaemia. Furthermore, respiratory failure increases the imbalance of oxygen supply and demand, ultimately leading to cardiopulmonary failure. In a recent prospective cohort study of 18 COVID-19 patients with elevated troponin levels, myocardial injury was assessed using cardiac magnetic resonance (CMR), echocardiography and endomyocardial biopsy (EMB). Structural and dynamic cardiac changes observed in this population included a mildly reduced median left ventricular ejection fraction of 52.5% [46.5–60.5%], moderately to severely reduced left ventricular global longitudinal strain of −11, 2% [−7.6% to −15.1%], myocardial tissue damage in 16 patients (83.3%) and myocardial oedema in 7 patients (38.9%) on magnetic resonance imaging, and lymphocytic myocarditis in 1/5 biopsied patients [12]. This observation supports the hypothesis of functional and morphological changes following structural damage of cardiomyocytes initially indicated by elevated troponin levels. 

An additional analysis of biomarkers of cardiac stress aside cardiac troponin B-type showed elevation of natriuretic peptide (BNP) in patients with cardiac injury hospitalized for COVID-19 [13]. Both biomarkers were correlated with illness severity and increased mortality. The hypothesis of cardiac injury related to COVID-19 is further supported by a recent retrospective observational study in 416 patients hospitalized for COVID-19 in Wuhan, China, which detected cardiac injury in 19.7% of patients. In this study, cardiac injury was associated with a higher risk of in-hospital mortality (42 of 82 [51.2%] vs. 15 of 334 [4.5%]; *p*  < 0.001) [6]. However, reported frequency of cardiac damage in COVID-19 patients shows high heterogeneity with incidences varying from 8% to 46.3% [5,9,14].

### 3.1. Direct Mechanisms of Cardiac Injury in COVID-19

Possible pathomechanisms of cardiac injury associated with COVID-19 affect the heart via direct and indirect pathways [4]. Direct mechanisms have been identified in several studies. An in vitro study demonstrated that SARS-CoV-2 infects cardiomyocytes via a cathepsin and angiotensin-converting enzyme 2 (ACE2) in a dependent manner [15]. Another experiment using artificially produced human capillary organoids was able to show that SARS-CoV-2 virus directly infects blood vessel cells [16]. At the cellular level, it is therefore possible that direct infection occurs in both cardiomyocytes and endothelial cells. A study in 39 autopsy cases of patients with COVID-19 confirmed an infection with SARS-CoV-2 in myocardial tissue, with highest activity in interstitial cells in addition to invading macrophages [17]. The presence of inflammation ultimately determines tissue damage.

According to a histological analysis in COVID-19 patients, recruitment of immune cells leads to endothelial dysfunction in several organs in which the ACE receptor is expressed including lung, heart, kidney and intestine. This in turn causes apoptosis and pyroptosis and thereby structural damage [18]. Furthermore, an autopsy study, comparing seven lungs of COVID-19 infected patients with lungs of patients who died from acute respiratory distress syndrome (ARDS) secondary to influenza A(H1N1) infection, revealed that endothelial damage and ruptured cell membranes as well as alveolar capillary microthrombi occur nine times more frequently in patients with COVID-19 than in patients with influenza [19]. Interestingly, the extent of new angiogenesis (“intussusceptive angiogenesis”) was 2.7 times higher than in the lungs of patients with influenza A(H1N1) infection [19]. Therefore, in addition to the initial predominant inflammatory reaction, there seem to be distinctive vascular features of severe endothelial injury in COVID-19 disease that differ from those in influenza infection.

### 3.2. Indirect Mechanisms of Cardiac Injury in COVID-19

Myocardial damage in patients with COVID-19 may also result from indirect mechanisms. These include conditions resulting in increased myocardial demand (tachycardia, hypotension, e.g., in sepsis, hypoxemia) and cardiac dysfunction (arrythmias, myocardial infarction), vascular thrombogenic structural changes (acute atherothrombosis, microthrombi) as well as a stress-induced cardiomyopathy (Takotsubo syndrome). In a retrospective analysis of 138 hospitalized patients with SARS-CoV-2 pneumonia in Wuhan, 17% developed cardiac arrhythmias (*n* = 23 of 138 total) among other cardiac complications and 16 patients (44%) required intensive care [20]. In fact, a variety of dysrhythmias have been observed in patients with COVID-19, including tachyarrhythmias, atrial fibrillation, flutter, ventricular arrhythmias, sustained ventricular tachycardia, ventricular fibrillation and atrial or ventricular ectopy as well as sinus tachycardia [21]. The possible causes of arrhythmogenicity in the context of COVID-19 include myocardial injury, hypoxia, systemic inflammation as well as therapy with QT prolonging drugs, drug interactions as well as common risk factors of arrhythmia, such as electrolyte abnormalities and cardiovascular comorbidities [22].

Another major pathophysiological aspect in the development of cardiac injury when infected with SARS-CoV-2 is a thrombotic milieu causing myocardial infarction among other thrombotic complications such as acute pulmonary embolism, deep-vein thrombosis, ischemic stroke and myocardial infarction [23]. The incidence of thrombotic complications in patients with COVID-19 is relatively high, ranging from 5% to 10% of critically ill patients [24], and amounting to up to 43% in those admitted at ICU [25]. The thrombotic milieu leads to a (hyper-)coagulopathic state that is linked to impaired microvascular perfusion and consecutive risk of myocardial infarction. In addition, atherothrombosis due to plaque instability and inflammatory endothelitis might contribute to increased risk of thromboembolic events [26].

Furthermore, cases of Takotsubo cardiomyopathy were reported in patients with COVID-19. In a recent report, a 72-year-old SARS-CoV-2 positive patient with cardiovascular risk factors and a history of recent stroke was diagnosed with Takotsubo cardiomyopathy [27]. The authors discussed a possible association of an increase in catecholamine levels induced by COVID-19 and recent acute stroke [27]. Takotsubo syndrome (TTS) is a stress-induced cardiomyopathy and has been associated with two causative conditions in the context of COVID-19, i.e., a direct complication of the infection and an indirect consequence of general psychological stress (such as social isolation and quarantine) [28]. In concordance with these hypotheses, TTS has also been reported in other infectious diseases, such as influenza virus and (bacterial) sepsis [29]. In addition to the usual stress factors and physical stressors related to systemic inflammatory diseases, isolation in quarantine and consecutive psychological distress might be additional stressors that are more specific for COVID-19. This is consistent with the observation that prevalence of mental health problems in the general population showed a substantial increase during the COVID-19 pandemic [30].

In addition, critical illnesses such as strokes, encephalitis/meningitis or seizures are potential physical stressors linked with TTS [31]. Further, in COVID-19, disease activation of the autonomic or “extended autonomic system” (EAS) may play an important role in developing TTS [32].

Figure 1 depicts factors and mechanisms whereby infection with SARS-CoV-2 might induce structural cardiac damage.

## 4. Pathophysiology and Clinical Manifestations of AIS in COVID-19

The frequency of stroke among COVID-19 patients shows heterogeneity in the literature with incidences up to 6% in cohort studies [33]. The inflammatory milieu is an integral part of stroke in COVID-19 patients. In a retrospective case series of 214 patients with COVID-19, neurologic events such as stroke were linked with a more severe infection, which may contribute to a higher mortality rate [34]. Importantly, incidence of stroke in COVID-19 seems to exceed a degree that can be explained by virus-mediated systemic inflammation alone. A retrospective cohort study of patients from two hospitals in New York (*n* = 1916) identified a higher rate of stroke patients with COVID-19 than in those with influenza infection (1.6% vs. 0.2%) [35]. These potentially known and additional mechanisms in COVID-19 disease will be discussed below. The interplay of mechanisms between cardiac injury and ischaemic stroke in COVID-19 is illustrated in Figure 2.

### 4.1. The Interplay of Coagulpathy and Inflammation in COVID-19 Related Stroke

Coagulopathy and cytokine storm are important factors in cardiovascular but also in cerebrovascular complications of COVID-19. Coagulopathy has been associated with the severity of COVID-19 disease [36]. Most case reports (*n* = 29) identified a large vessel occlusion, the middle artery and the anterior circulation being most common [37]. Large artery occlusion mediated strokes were connected with the hypercoagulable state in some case reports [38,39,40] The coagulopathy associated with COVID-19 ultimately underlies the known pathophysiological mechanisms, the Virchow triad—endothelial cell damage, abnormal blood flow dynamics and platelet activation [41]. Interestingly, in a retrospective cohort study of 3556 hospitalized patients with COVID-19 (ischemic stroke occurring in 0.9%) cryptogenic stroke was the most frequent stroke etiology found (65.6%) [42]. However, it needs to be acknowledged that the applied TOAST classification of stroke etiology does not differentiate cryptogenic stroke from embolic stroke of undetermined source, the latter being frequently linked to cardiac embolism. The pathophysiological characteristics of COVID-19-related stroke are yet to be defined. Neuropathological studies revealed vascular congestion, micro-thrombotic infarction and microhaemorrhage in addition to hypoxic-ischemic changes [37]. The autopsy of a patient infected with SARS-CoV-2 revealed the presence of virus-like particles in the cytoplasm of a neuronal cell body (localized in the frontal lobe in this particular case) as well as in capillary endothelial cells—possibly enabling entry in CNS via a hematogenous route [43]. In this case report, the authors proposed that the virus may infect endothelial cells of the blood–brain barrier or epithel cells of the blood–cerebrospinal fluid barrier [43,44]. The primary haematogenous pathway would also be consistent with an autopsy study in brain tissue that found ACE2 expressed solely in endothelium and vascular smooth muscle cells [45]. However, there are several investigations that promote a neural pathway whereby SARS-CoV-2 pathology propagates to the brain as well. Among postulates, a neuronal route via afferent olfactory nerves and transsynaptic spread (endo-/exocytosis) has been frequently discussed [44,46]. This might be a complementary mechanism contributing to stroke risk by inducing brain damage and vulnerability in COVID-19 patients. After entry into the CNS, systemic inflammatory induced pathomechanisms of COVID-19 also play a role in the development of ischaemic stroke. Following activation of immune system after infection, a release of proinflammatory factors (such as cytokines, e.g., IL-6) consecutively activates the coagulation cascade, resulting in fibrin deposition and platelet activation and ultimately microthrombus formation [37]. In presence of a diffuse endothelial dysfunction, the balance of the coagulation process, including the natural formation of anticoagulants, such as antithrombin, protein C and tissue factor pathway inhibitor (TFPI), is disturbed [47]. Furthermore, an acute fibrinolytic reaction (induced by tissue plasminogen activator (t-PA) and urokinase-type plasminogen activator (u-PA)) results in the formation of plasmin, which breaks down fibrin into fibrin degradation products—leading to an increase in D-dimers [37].

### 4.2. Role of the ACE2 Receptor in Stroke Related to COVID-19

Downregulation of the ACE2 receptor seems to be involved in the stimulation of inflammation, coagulation and endothelial dysfunction [37]. The ACE2 receptor causes cleavage and thus degradation of angiotensin II. When the ACE2 receptor is downregulated, angiotensin II, which is known as a vasoconstrictor, increases and causes hypertension. This can lead to rupture of aneurysms consequently causing subarachnoid haemorrhage, and—in the longer term—become a risk factor for stroke via endothelial damage. Angiotensin II acts as a prothrombotic mediator (e.g., activation of macrophages and other immune cells) and inhibits fibrinolysis (by increasing Plasminogen activator inhibitor-1 (PAI-1) production) [48]. Both the expression level and pattern of ACE2 in different tissues, are thought to be important for susceptibility and therapeutic response in patients with COVID-19 [49]. Thus, the Angiotensin-converting enzyme (ACE)-Angiotensin II (Ang II)-Angiotensin II type 1 receptor (AT1R) axis seems to be involved in worsening of hypertension, atherosclerosis and thrombogenesis [50]. In contrast, ACE2 is converted to angiotensin, which is neuroprotective. Binding the Mas receptor, angiotensin has positive effects on blood pressure (via nitric oxide and bradykinin), progression of atherosclerosis, antithrombotic effects and positive effects on infarct size via antioxidant and anti-inflammatory effects [50]. Under normal conditions there is an interaction of the axes with the ACE2→Angiotensin1–7→Mas receptor axis balancing and weakening the (adverse) ACE→Angiotensin II→AT1 receptor axis [37]. Ultimately, after SARS-CoV-2 infection, this balance is dysregulated [51].

### 4.3. Laboratory Markers in Stroke Related to COVID-19

The clinical value of laboratory chemical markers is closely related to the pathogenesis of COVID-19. In the context of coagulopathy elevated D-dimer levels (≥2-fold above the normal range), higher lactate dehydrogenase levels, prolonged prothrombin time, thrombocytopenia and decreased fibrinogen levels (especially with consumption coagulopathy that occurs with disseminated intravascular coagulation (DIC)) have been reported [36,52]. In addition, the prevalence of antiphospholipid antibodies in critically ill patients was high (85% to 87.7%) compared to other viral or bacterial infections [53,54]. It has been demonstrated that anti-phospholipid antibodies, including lupus anticoagulants (which are also associated with anti-phospholipid syndrome (APS)) increase thromboembolic risk in both arterial and venous system [55]. Furthermore, the presence of antiphospholipid antibodies has been associated with stroke [56]. So far, elevated levels of antiphospholipid antibodies have been particularly associated with severe cases of COVID-19 [57,58]. In a case series of patients with COVID-19 hospitalized for stroke, in 5 of 6 patients positive antiphospholid antibodies (Anticardiolipin (aCL), Anti-β2-glycoprotein-1 (aβ2GPI), Lupus anticoagulant) were noted [58]. However, a prospective observational study confirmed the high prevalence of antiphospholipid antibodies in COVID-19, but was not able to link this to the occurrence of thrombosis [53]. Severe cases of COVID-19 were associated with increased platelet activation in laboratory tests (plated surface expression of CD62P (P-selectin) and CD63), while mildly affected or asymptomatic patients did not show this association [59]. In addition, ex vivo experiments showed that the inflammatory milieu in severe COVID-19 disease contributes to platelet activation [59]. Furthermore, neutrophil extracellular traps (NETs), networks of chromatin, microbicidal proteins and oxidizing enzymes released by neutrophils, can promote microvascular thrombosis [60]. Hypercoagulation, as well as endothelial activation and infection, contribute to coronary microvascular dysfunction [61]. Perhaps, microvascular dysfunction is a general phenomenon in COVID-19 patients that includes cerebral vessels, but this assumption remains speculative.

### 4.4. The Role of SARS-CoV-2 Related Cardiac Injury in COVID-19 Patients with AIS

Structural changes occur due to the SARS-CoV-2 infection connecting the cardiopulmonary with the cerebral vascular system. The question arises, whether pre-existing cardiac damage is relevant to the consequences of COVID-19 disease. In a large retrospective cohort study including 1162 patients with acute coronary syndrome (ACS), reduced LVEF was associated with increased susceptibility to COVID-19 [62]. However, in a multicenter cohort study with 305 COVID-19 patients, cardiac structural abnormalities were detected in nearly two-thirds of the patients receiving transthoracic echocardiography—contributing to a higher mortality rate [63]. In addition, commonly known risk factors may further increase the co-occurrence of cardiac disease and ischaemic stroke. Risk factors such as hypertension, dyslipidemia and diabetes are associated with a pathological downregulation in ACE2 [64]. An Italian study characterized the baseline statistics of patients with COVID-19 (*n* = 1591) [65]. They found, that COVID-19 patients had an increased rate of common comorbidities, such as hypertension in 49% [95% CI, 46–52%], cardiovascular disease in 21% [95% CI, 19–24%] and hypercholesterolemia in 18% [95% CI, 16–20%] of cases [65]. In a study using image-based AI to characterize the tissues of a COVID-19 patient and classify the severity of infection, multiple pathways of cardiac and brain involvement in COVID-19 infection were detected [66].

In the so-called “hypoxia pathway”, the primary cause is a respiratory infection, which can lead to ARDS. This respiratory hypoxia can ultimately lead to both myocardial ischaemia and hypoxia in the brain [66]. In consequence, hypoxemia and the increase in CO2 in the brain leads to cerebral vasodilation and oedema. Another unifying concept is the endothelial dysfunction, which is triggered, among other things, by deregulation of the RAAS system [66] The third overlapping pathway is the immune response to the SARS-CoV-2 virus, which can lead to an increase in inflammatory parameters with possible cytokine storm, causing plaque rupture affecting the heart and brain vessels [66]. Infection with SARS-CoV-2 thus induces key changes including immune response, hypoxia and endothelial dysfunction and multiple overlapping pathways may ultimately lead to cerebral involvement including cerebral ischaemia.

## 5. Discussion

Pathomechanisms of COVID-19, including downregulation of ACE2, the inflammatory and thrombotic milieu as well as respiratory dysfunction seem to be connected with cardiac injury and risk of ischaemic stroke on multiple levels.

In general, neurological manifestations in COVID-19 are highly variable (including headache, impaired consciousness, ataxia, tremor, meningitis, encephalitis, cerebral haemorrhage, subarachnoid haemorrhage, seizures) and are associated with different MRI findings involving ischemic lesions [67]. It is conceivable that the number of unreported stroke patients with COVID-19 infection is higher due to the availability and increased expense of imaging as well as possible “silent” clinical presentation. Finally, in a review, it was found that about 17.85% of patients who underwent neuroimaging had ischaemic changes suggestive for a stroke [68]. The inflammatory as well as the thrombogenic milieu seems to be a joint condition of cardiac and cerebrovascular injury in COVID-19 patients. Common patterns of damage including primary endothelial damage as well as plaque rupture as a result of inflammation. Other coexisting pathways include microvascular dysfunction, possibly contributing to ischaemic lesions in functionally relevant areas, a “last meadow”. Coronary microvascular dysfunction is discussed to cause the Takotsubo syndrome [69]. In the case of large vessel occlusion, the focus is on potential sources of embolism, which in the case of COVID-19 related stroke frequently seems to be of cardiac origin. The outlined structural changes and rhythm disturbances could lead to the development of thrombi. In the case of COVID-19 disease, concomitant hypercoagulopathy could be a key contributor. In addition, existing cardiovascular risk factors such as diabetes, arterial hypertension, a lipid metabolism disorder or nicotine excess can additionally increase the risk of cardiovascular diseases including stroke. As pointed out recently, care of stroke patients is challenging during the COVID-19 pandemic not only because a past history of stroke is a risk factor of severe COVID-19 whereas severe COVID-19 is a risk factor of stroke, forming a vicious circle on a pathophysiological level, but also because infection control measures may impact care of stroke patients [70,71,72,73].

Understanding the multiple pathophysiological links to cardio- and cerebrovascular diseases in COVID-19 might help optimize cardioprotection and neuroprotection in these patients. For instance, statins have, among others, pleiotropic anti-inflammatory effects and may attenuate endothelial dysfunction via upregulation of ACE. The potential benefit of these features in COVID-19 patients have been discussed [74]. The involvement of the renin-angiotensin system (RAS), an interaction with cholinergic neurotransmission via neuronal nicotinic acetylcholine receptors (nAChR) has also been proposed as a potential treatment target in COVID-19 [75,76]. While multiple hypotheses in cardioprotection and neuroprotection in COVID-19 exist, further studies are needed to understand the connecting pathomechanisms and facilitate robust data on effective treatment of COVID-19 patients with cardiac injury and those with AIS.

## 6. Conclusions

The COVID-19 pandemic has presented new challenges to medicine. Ultimately, understanding the individual pathomechanisms of the disease may be crucial in preventing cardiac and cerebral damage in those patients at risk.

## Figures and Tables

**Figure 1 life-12-00075-f001:**
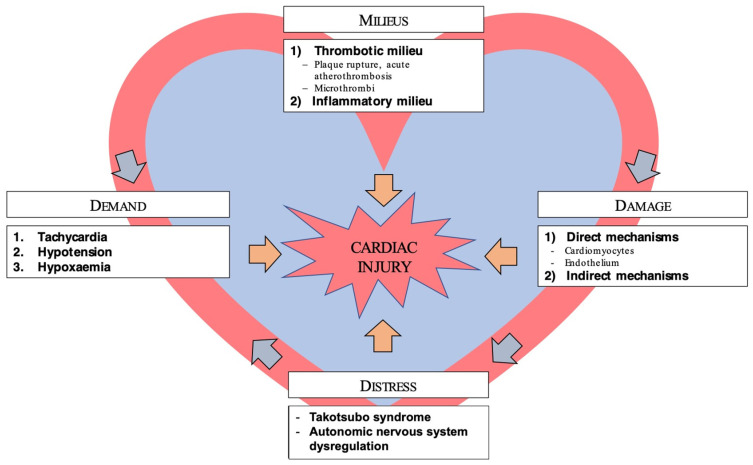
Mechanisms of cardiac injury in COVID-19.

**Figure 2 life-12-00075-f002:**
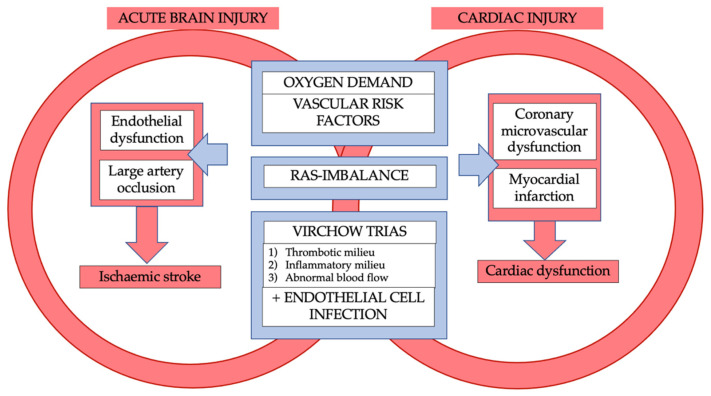
Interplay of mechanisms between cardiac injury and ischaemic stroke in COVID-19.

## Data Availability

Not applicable.

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
