# Peer review of "Pathophysiology of Cardiac Injury in COVID-19 Patients with Acute Ischaemic Stroke: What Do We Know So Far?—A Review of the Current Literature"

_life, 2022, doi:10.3390/life12010075_

Round 1
Reviewer 1 Report
I enjoyed reviewing your article. It is timely and would be greatly appreciated
Author Response
Dear Reviewer,
thank you very much for your rating and appreciation on our review.
Best regards,
Daniela Schöne
Reviewer 2 Report
This manuscript reviewed the recent epidemiological and researching progress of cardiac injury in the COVID-19 cohort, and summarized pathophysiology and clinical manifestations of cardiac injury in COVID-19. More importantly, in terms of the pathological consequences of cardiac injury and internal milieu features of cardiac injury patients with COVID-19, e.g. inflammatory, coagulation and endothelial dysfunction, the author suggest that cardiac injury accompanied with COVID-19 might be an underlying reason causing higher ischemic incidence, and described the potential mechanism. Finally, the potential strategies preventing ischemic occurring and promoting outcome after ischemia were also generally discussed. In general, this is an interesting, focused and well-designed review paper. The following points needs to be addressed before further decision by journal editor.
- Typo: there is no subtitle for section 4.
- Suggest preparing a schematic diagram showing the potential interplay and signaling between heart injury and ischemic stroke in the COVID-19 cohort. It could include the author’s assumption, evidences and others. It could be put in section 4.
Author Response
Dear reviewer,
thank you very much for your comments on our review. At your suggestion, the following changes were made:
- Subtitle for section 4 was hidden because of overlapping with the figure.
- Schematic diagram showing the potential interplay and signaling between heart injury and ischemic stroke in the COVID-19 cohort.
Best regards,
Daniela Schöne
Reviewer 3 Report
The paper presented “multiple pathways” of stroke/cardiac-related COVID-19 damage, including hypoxia, immune response, and endothelial dysfunction pathways but failed to expand appropriately on any of them beyond citing scattered clinical findings. In other words, the pathophysiology was not thoroughly discussed as promised, and mechanisms were merely mentioned as opposed to explored. For example, it was mentioned that inflammation alone cannot explain insults to the brain (line 202): why is this and what further explanations are there that may help elucidate brain injury (the three multiple pathways were mentioned, but again, they were not elaborated upon)?
In addition to the lack of detail on the pathophysiology of COVID-19 related cardiac and neural insults, clinical data could have been analyzed as opposed to merely regurgitated. For example, the paper mentioned that COVID-19 patients with cardiac injury have increased mortality (lines 35, 107) but failed to mention the magnitude of the increase. Providing an odds ratio would have been appropriate here.
Also, the prose, especially the English grammar and spelling mistakes, seemed careless and did not inspire confidence in the reader.
Author Response
Dear reviewer,
thank you very much for your comments on our review. With the present review we want to give an overview of the individual mechanisms, which have already been discussed in detail in other articles – focusing on the connection between heart and brain in COVID 19 patients. The particularly involved mechanisms were discussed offering future targets for prevention research. Moreover, the establishing of biomarkers in this clinical context is valuable. The intention of the review is primarily to raise awareness on this topic.
Line 202: Corrected argumentation – heterogeneity in frequency of stroke, inflammatory milieu as integral part of stroke in COVID-19 patients, BUT higher incidence of stroke compared to other virus infections supports the hypothesis that other mechanisms underlying the COVID19 disease are involved.
Line 35, 107: Clinical data for increased mortality in COVID-19 patients with cardiac injury were added.
The text was revised for mistakes in English language and style.
Best regards,
Daniela Schöne
Reviewer 4 Report
MAJOR COMMENTS:
The authors mention that this is a narrative review. They have not stated how many articles they found from their search, and how they decided to include/exclude studies. Is there a selection bias in how they selected studies for inclusion in this narrative review?
Line 254 mentions rupture of aneurysms. Could the authors briefly discuss if there a link with rupture of intracerebral aneurysms and haemorrhagic strokes (as opposed to ischaemic strokes)?
MINOR COMMENTS:
Line 33: do (7,8,9) indicate references?
Line 57: "inmortality" should be "in mortality"
Line 60: explain abbreviation "PCT" - is this procalcitonin?
Line 61: "mechanismus" - do authors mean "mechanisms"?
Line 71: "includedstudies" should be "included studies"?
Line 95: also state specifically that brackets indicate "interquartile range"?
Line 96: also state "median" global longitudinal strain?
Line 115: should "cathepsins" be "cathepsin"?
Line 116: "ahow" should be "show"?
Lines 140 & 163 "Takotsubo" but lines 161 & 165 & 343 & in figure 1 "Tako-Tsubo" - please make consistent throughout manuscript.
Line 166: "cardiopathy" should be "cardiomyopathy"
Line 173: "COVI-19" should be "COVID-19"
Line 178: should EAS stand for "extended autonomic system"?
Line 209: "coagulopathy" should be "coagulopathy"
Line 257: please expand "PAI-1" abbreviation
Line 276: Why are there two numbers in brackets "(85%; 87.7%)" - is this 95% CI or range or something else? If 95% CI then should include mean?
Line 317: hyphen does not seem necessary?
Line 349: I think the authors mean "excess" rather than "abscess"?
Line 359: "this" should be "these"?
Figure 1: "MILIEUS" or "MILIEU"?
Author Response
Dear reviewer,
thank you very much for your comments on our review. At your suggestion, the following changes were made:
- The following text has been added (line 79):
The initial search provided 19,090 results, which were screened according to content aspects related to the topic of the review and evaluated according to relevance. Explicit exclusion criteria were not defined. A selection bias cannot be ruled out.
- Linie 269: This can lead to rupture of aneurysms consequently causing subarachnoid hemorrhage, and - in the longer term - become a risk factor for stroke via endothelial damage.
Minor comments:
Line 33: do (7,8,9) indicate references?
- Yes, were transformed.
Line 57: "inmortality" should be "in mortality"
- Corrected.
Line 60: explain abbreviation "PCT" - is this procalcitonin?
- PCT = Procalcitonin. Corrected.
Line 61: "mechanismus" - do authors mean "mechanisms"?
- Corrected.
Line 71: "includedstudies" should be "included studies"?
- Corrected.
Line 95: also state specifically that brackets indicate "interquartile range"?
- Corrected (In original article normal brackets)
Line 96: also state "median" global longitudinal strain?
- Original article “Median LVEF on echocardiography was 52.5% (46.5%–60.5%).”
Line 115: should "cathepsins" be "cathepsin"?
- Corrected.
Line 116: "ahow" should be "show"?
- Corrected.
Lines 140 & 163 "Takotsubo" but lines 161 & 165 & 343 & in figure 1 "Tako-Tsubo" - please make consistent throughout manuscript.
- Corrected, now consistent “Takotsubo”.
Line 166: "cardiopathy" should be "cardiomyopathy"
- Corrected.
Line 173: "COVI-19" should be "COVID-19"
- Corrected.
Line 178: should EAS stand for "extended autonomic system"?
- Corrected.
Line 209: "coagulopathy" should be "coagulopathy"
- I don't understand, this is the same word?
Line 257: please expand "PAI-1" abbreviation
- Corrected.
Line 276: Why are there two numbers in brackets "(85%; 87.7%)" - is this 95% CI or range or something else? If 95% CI then should include mean?
- This is the prevalence in the literature cited.
Line 317: hyphen does not seem necessary?
- Hyphen deleted.
Line 349: I think the authors mean "excess" rather than "abscess"?
- Corrected.
Line 359: "this" should be "these"?
- Corrected.
Figure 1: "MILIEUS" or "MILIEU"?
- Milieus as the plural noun (thombotic and inflammatory mileu)
Best regards,
Daniela Schöne
Round 2
Reviewer 3 Report
This reviewer paper does not provide any valuable perspective insight about COVID-19 in cardiac injury and stroke, and whatever information covered and discussed has been well known. The title of "Pathophysiology of cardiac injury in COVID-19 patients with acute ischaemic stroke" is very big, however, is not supported by the component of the paper covered.
Author Response
Dear reviewer,
we would like to thank you for your evaluation of our article. We entirely agree with your suggestion to improve the title of our manuscript. We changed the title to: „Pathophysiology of cardiac injury in COVID-19 patients with acute ischaemic stroke: what do we know so far? – A review of the current literature“ in order to undersocre the narrative nature of this review. In order to improve the illustration of the mechanisms presented in our review, we added a new figure depicting the possible interplay between heart and brain damage in COVID-19. This change is also corresponding to the comment of reviewer 2. In summary, we thank the reviewers for their helpful suggestions that have led to substantial improvements of our review.
Best regards,
Daniela Schöne
Reviewer 4 Report
Figure 1 still has 'Tako-Tsubo' which is inconsistent with 'Takotsubo' elsewhere.
Author Response
Dear reviewer,
thank you for your careful review - the text in figure 1 has been changed.
Best regards,
Daniela Schöne